# Publication bias in pharmacogenetics of adverse reaction to antiseizure drugs: An umbrella review and a meta-epidemiological study

**S. Bally[1], J. Cottin[2], M. C. Gagnieu[3], J. C. Lega[1,4], C. Verstuyft[5,6], S. Rheims[7], G. Lesca[8], M. Cucherat[1,2], Guillaume Grenet[2]***

**1** Laboratoire de Biométrie et Biologie Evolutive UMR5558, Université Lyon 1, CNRS, Villeurbanne, France, **2** Service Hospitalo-Universitaire de Pharmacotoxicologie, Pôle de Santé Publique, Hospices Civils de Lyon, Lyon, France, **3** Laboratoire de Pharmacologie, Groupement Hospitalier Sud, Hospices Civils De Lyon, Lyon, France, **4** Service de Médecine Interne et Vasculaire, Hôpital Lyon Sud, Hospices Civils de Lyon, Lyon, France, **5** CESP, MOODS Team, INSERM, Faculté de Médecine, Université Paris-Saclay, Le Kremlin Bicêtre, France, **6** Service de Génétique Moléculaire, Pharmacogénétique et Hormonologie de Bicêtre, Hôpitaux Universitaires Paris-Sud, Assistance Publique-Hôpitaux de Paris, Hôpital de Bicêtre, Le Kremlin Bicêtre, France, **7** Department of Functional Neurology and Epileptology, Hospices Civils de Lyon, Lyon 1 University, Lyon, France, **8** Service de Génétique, Groupement Hospitalier Est, Hospices Civils De Lyon, Université Lyon 1, Lyon, France

\* guillaume.grenet@chu-lyon.fr

**Data Availability Statement:** All relevant data are within the paper and its Supporting information files.

## Abstract

Publication bias may lead to a misestimation in the association between pharmacogenetic biomarkers (PGx) and antiseizure drug's adverse effects (AEs). We aimed to assess its prevalence in this field. We searched for systematic reviews assessing PGx of antiseizure drug's AEs. For each unique association between a PGx, a drug and its AE, we used the available odds ratio (ORs) to generate corresponding funnel plots. We estimated the prevalence of publication bias using visual inspections and asymmetry tests. We explored the impact of publication bias using ORs adjusted for potential publication bias. Twenty-two associations were available. Our visual analysis suggested a publication bias in five out twenty-two funnel plots (23% [95%CI: 8; 45]). The Egger's test showed a significant publication bias in one (HLA-B*15:02 and phenytoin-induced Stevens-Johnson syndrome or toxic epidermal necrolysis, p = 0.03) out of nine (11% [95%CI: 0; 48]) and the Begg's test in one (HLA-B*15:02 and carbamazepine-induced serious cutaneous reactions, p = 0.02) out of ten (10% [95%CI: 0; 45]) assessable funnel plots. Adjusting for publication bias may reduce by half the ORs of the pharmacogenetics associations. Publication bias in the pharmacogenetic of antiseizure drug's AEs is not uncommon and may affect the estimation of the effect of such biomarkers. When conducting pharmacogenetic studies, it is critical to publish also the negative one.

**Funding:** The authors received no specific funding for this work.

**Competing interests:** I have read the journal's policy and the authors of this manuscript have the following competing interests: SB, JC, MGC, GG declare that they have no competing interest. JCL has received speaker fees and honoraria from Roche. CV has received consulting fees from Genzyme, Novartis and speaker honoraria from Galapagos. SR received speaker and/or consulting fees from UCB Pharma, EISAI, GW Pharma, Idorsia, LivaNova, and Arvelle Therapeutics. GL has received speaker honoraria from GWpharma, Eisai and Biomarin. MC has received consulting fees from Boehringer Ingelheim, SANOFI, AstraZeneca, EISAI and speaker honoraria from SANOFI. This does not alter our adherence to PLOS ONE policies on sharing data and materials.

**Abbreviations:** ADRs, adverse drug reactions; AE, adverse effect; CI, Confidence Interval; HSS, hypersensitive syndrome; IdP, identification number; MPE, maculopapular exanthema; ND, not determinable; OR, OR obtained without adjustment for publication bias; ORs, odds ratio; OR$_{TM}$, OR estimated with the Trim and Fill method; PGx, pharmacogenetic biomarkers; pMA, published meta-analyses; PRISMA, Preferred Reporting Items for Systematic Reviews and Meta-Analyses; ROR, ratio of the ORs; SCRs, serious cutaneous reactions; SJS, Stevens-Johnson syndrome; T, treated tolerant patients; TEN, toxic epidermal necrolysis; TM, Trim and Fill method; U, untreated patients.

# Introduction

## Rationale

Systematic reviews and meta-analyses help synthesize the estimates from several clinical studies. However, their results may be affected by publication bias [1]. At the beginning, publication bias has been pointed out regarding the risk of treatment efficacy overestimation [2]. Then, publication bias has also been identified for assessing the risk of adverse drug reactions (ADRs) [3]. Furthermore, publication bias has been documented in genetic epidemiology in general [4]. Previous meta-epidemiological studies assessed publication bias in various areas [5–8]. However, the extent of publication bias in pharmacogenetics remains unclear.

Several genetic variants have been associated with an increased risk of antiseizure drug's adverse effects (AEs) [9]. Some of them seem to be reliable, as HLA-B and carbamazepine/phenytoin-induced severe cutaneous reactions, allowing clinical implementation of pharmacogenetic results [10, 11]. Others showed some discrepancies, such as for lamotrigine and the risk of Stevens-Johnson syndrome (SJS) or toxic epidermal necrolysis (TEN) [10]. Such pharmacogenetic associations may also vary across different populations [12]. A publication bias has been documented regarding the assessment of antiseizure drugs, such as topiramate [13] or pregabalin for example [14]. Previous meta-analyses assessing pharmacogenetic risk factors of antiseizure drug's AE suggested a potential publication bias, including HLA-B*15:02 polymorphism [15]. Other meta-analyses did not report a publication bias, but the number of available trials was often low, ranging from two to eleven studies in three meta-analysis [16–18]. Thus, the statistical power for detecting funnel plot asymmetry was probably insufficient [19]. The prevalence and the potential impact of publication bias in the pharmacogenetics of antiseizure drug's AEs remain unclear.

## Objectives

Our hypothesis was that the publication bias is particularly sizable in the pharmacogenetics of antiseizure druginduced adverse reactions. We aimed to assess its prevalence in this area first. Then, we aimed to illustrate its potential impact on the estimation of those pharmacogenetics biomarkers.

# Methods

## Protocol

We conducted a meta-epidemiological study. We did not register the protocol, but we formulated the hypothesis *a priori*. Methods and results were reported following the Preferred Reporting Items for Systematic Reviews and Meta-Analyses (PRISMA) statement for systematic review [20] and the dedicated guideline for reporting meta-epidemiological methodology research [21]. We conducted an umbrella review to identify systematic reviews and meta-analyses of interest. We did not search directly for clinical studies themselves. Indeed, the assessment of publication bias requires as many clinical studies as possible, and is expected to be inconclusive in case of isolated (i.e non-replicated) clinical studies. Therefore, we opted for an umbrella review of published systematic reviews. One bibliographic reference of a systematic review could include several published meta-analyses (pMA), one for each unique triplet association [genotype-drug-ADR]. Each unique triplet association [genotype-drug-ADR] may be studied across different published meta-analyses. We gathered all the informative clinical studies assessing the same triplet association [genotype-drug-ADR] reported across the published meta-analyses from the different systematic reviews. Combining the data from previous overlapping published meta-analyses allowed increasing sample size of data given the number of clinical

studies. We then assessed the presence of a publication bias and its impact for each unique triplet association [genotype-drug-ADR].

## Eligibility criteria

Systematic reviews were included if they met the following inclusion criteria: (i) meta-analyses of clinical studies (observational or comparative trials) addressing antiseizure drugs and (ii) reporting of ADR related to pharmacogenetic biomarkers.

## Information sources

We conducted an umbrella review on PubMed, up to January 29, 2019, seeking published meta-analyses investigating the association between a genetic variant and an adverse reaction in patients treated with antiseizure drugs. We limited the information source to the Medline database. Indeed, the Clinical Pharmacogenetics Implementation Consortium (CPIC) Guideline for HLA genotype and Use of Carbamazepine and Oxcarbazepine was restricted to the use of "the PubMed database" [11]. Using the same restriction allowed to estimate the impact of publication bias in the context of the current practice of guidelines elaboration in this area.

## Search strategy

The Medline database was searched using the following keywords: "pharmacogen*" OR "genetic variant" OR "polymorphism" OR "allele" OR "association" AND "epilepsy" OR "antiepileptics" OR "anticonvulsants" AND "adverse event" OR "toxicity" OR "serious event" OR "meta-analysis" OR "systematic review" (see the search strategy in Supplementary information), without restrictions regarding the year of publication.

## Study selection and data extraction

Abstracts of bibliographic references were screened based on their title and abstract, and then selected using their full text, reason for exclusion being tracked. We extracted descriptive characteristics: authors, publication date, type of study, and ethnicity. We extracted study design, pharmacogenetic associations as reported by the authors (name of the genetic variant, antiseizure drug assessed, control treatment used, and adverse reactions). Indeed, as we used previously published systematic reviews of clinical studies, our extraction is limited by the author's definition of the genetic variant (i.e not the rs number if not reported) and of the adverse drug reactions. We extracted the odds ratios (OR) with their 95% confidence intervals (95%CI) of each clinical studiy reported in the included published meta-analyses.

## Summary measures

We used odds ratio (OR) and its 95% confidence interval (95%CI) to estimate the association between the genotype and the risk of the drug induced adverse reaction.

## Synthesis of result–data analysis

For each unique triplet association [genotype-drug-ADR], we gathered the ORs of the clinical studies from different published meta-analyses. We limited the compilation to comparisons using treated patients as control groups.

We used the funnel plot approach for assessing potential publication bias [22, 23]. We generated the funnel plots for each included associations, for estimating the prevalence of publication bias. Firstly, we conducted a visual analysis of the generated funnel plots [24]. Two researcher independently assessed if a publication bias was 'likely', 'unlikely', or 'not

determinable' (SB, GG). Agreement was estimated using a Free-marginal kappa estimator [25]. A third researcher helped resolve disagreements, blinded to the previous diagnoses (JC). Secondly, asymmetry was tested using the Egger [26] and Begg's methods [27] (function meta-bias, package {meta}) if at least five studies were available. P-value <0.05 were considered significant without adjustment for multiple testing. We calculated and reported the proportion of publication bias according to these three methods. Given the guidance of Cochrane handbook, the fail-safe number method was not used [22].

For exploring the potential impact of publication bias, we used the Trim and Fill [28] method to adjust the OR for potential publication bias [29]. We applied the Trim and Fill function (estimator L, fixed-effect model) if more than five estimates and their 95%CI were available for the same association $_{[genotype-drug-ADR]}$). Then, we compared the OR obtained without adjustment for publication bias ($OR_{NP}$) and the OR estimated with the Trim and Fill method ($OR_{TM}$). We also tested for an interaction between each pair of ORNP−$OR_{TM}$, using the ratio of the ORs (ROR) and its 95%CI. We conducted the analyses on R 3.3.1 [30] (package {meta}, version 4.9–4) [31].

## Results

### Study selection

From 295 references identified on PubMed, we included ten systematic reviews (see Fig 1). From them, we removed one systematic review [32] that contained estimates with infinite confidence intervals, limiting their use in our study. The nine usable systematic reviews included 33 published meta-analyses [15–18, 33–37]. Among them, 22 unique triplet associations $_{[genotype-drug-ADR]}$ were available for analysis.

### Study characteristics

The characteristics of included meta-analyses (published meta-analyses and each specific triplet association $_{[genotype-drug-ADR]}$) are detailed in the Table 1. In addition to the first author with the corresponding bibliographic reference of the published systematic review, a specific 'identification number' identified each published meta-analysis, a specific letter identified each specific triplet association $_{[genotype-drug-ADR]}$. Most of the included patients were Asian. Most of the gene variants were related to the HLA system. One published meta-analysis addressed the genetic polymorphisms of CYP2C9. Six antiseizure drugs were assessed (carbamazepine, lamotrigine, levetiracetam, phenobarbital, phenytoin, and valproate). The reported ADR were: "hypersensitivity", hypersensitive syndrome (HSS), maculopapular exanthema (MPE), serious cutaneous reactions (SCRs), SJS, and TEN. All the selected meta-analyses included case-control study design. Used comparators were: treated but tolerant patients in 28 published meta-analyses, untreated patients in four published meta-analyses, and both in one published meta-analysis.

### Prevalence of the publication bias

Twenty-two funnel plots of specific associations $_{[genotype-drug-ADR]}$ were generated. The funnels plots are available in the S1 Fig in S1 File. The Table 2 summarises the estimation of the prevalence of publication bias.

**Visual diagnosis of funnel plots.** For the visual analysis of the funnel plots, before reaching consensus with the third reviewer, the percentage of overall agreement between the two initial reviewers was 64%, Free-marginal kappa = 0.45 [95%CI: 0.15; 0.76]. Our visual analysis of generated funnel plot estimated that a publication bias was i) "likely" in five (23% [95%CI:

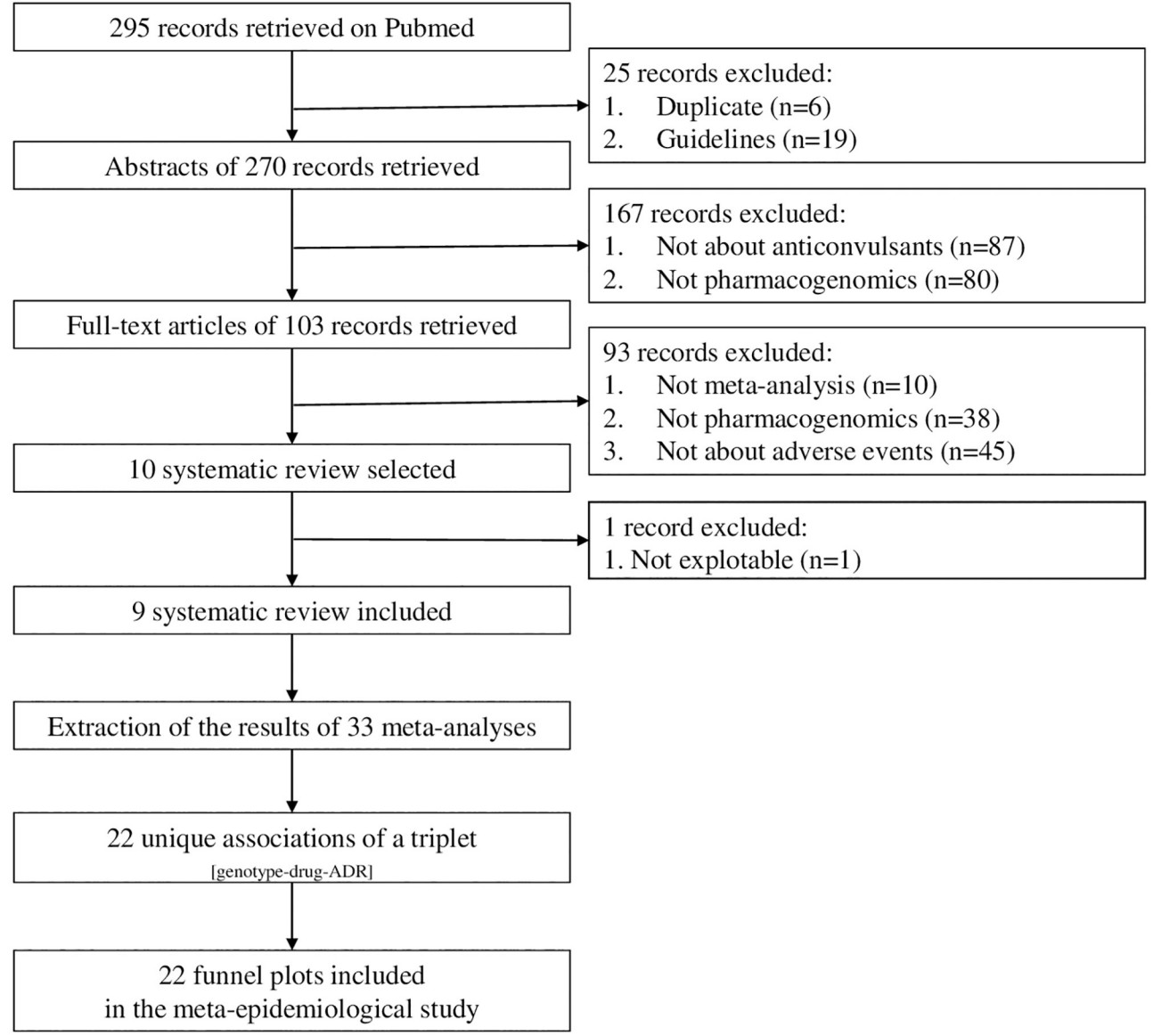

**Fig 1. Flow diagram of the bibliographic search.**

8; 45]) out of 22 exploitable funnel plots; ii) "unlikely" in 3 (14% [95%CI: 3; 35]) out of 22 exploitable funnel plots, and iii) "not determinable" in 14 (64% [95%CI: 41; 83]) out of 22 exploitable funnel plots.

**Assymetry tests.** The Egger's test i) showed a significant ($p<0.05$) publication bias in one out of nine assessable funnel plots (11% [95%CI: 0; 48]). It was not able to conclude ($p>0.05$) in eight out of nine assessable funnel plots (89% [95%CI: 52; 100]). The funnel plots were not exploitable in 13 out of 22 cases (59% [95%CI: 36; 79]).

The Begg's test showed a significant ($p<0.05$) publication bias in one out of 10 assessable funnel plots (10% [95%CI: 0; 45]). It was not able to conclude ($p>0.05$) in 9 out of 10 assessable funnel plots (90% [95%CI: 55; 100]). It was not assessable in 12 out of 22 exploitable funnel plots (55% [95%CI: 32; 76]).

**Table 1. Characteristics of included meta-analyses.**

| Author | IdP | Genotype | Antiseizure drug | ADR | Control | Ethnicity | Association |
|---|---|---|---|---|---|---|---|
| Chouchi [15] | 1 | HLA-B*15:02 | Carbamazepine | SCRs | T | Mostly Asian | A |
| Chouchi [15] | 2 | HLA-B*15:02 | Carbamazepine | SJS | T | Mostly Asian | B |
| Chouchi [15] | 3 | HLA-B*15:02 | Carbamazepine | SJS, TEN | T | Mostly Asian | C |
| Grover [16] | 10 | HLA-B*15:02 | Carbamazepine | SJS, TEN | T | Asian | C |
| Tangamornsuksan [34] | 23 | HLA-B*15:02 | Carbamazepine | SJS, TEN | U&T | Asian | C |
| Yip [37] | 31 | HLA-B*15:02 | Carbamazepine | SJS, TEN | T | Asian | C |
| Deng [33] | 4 | HLA-B*15:02 | Lamotrigine | SJS, TEN | T | Asian | D |
| Grover [16] | 17 | HLA-B*15:02 | Lamotrigine | SJS, TEN | T | Asian | D |
| Li [18] | 22 | HLA-B*15:02 | Lamotrigine | SJS, TEN | T | Asian | D |
| Zeng et al [17] | 33 | HLA-B*15:02 | Lamotrigine | SJS, TEN | T | Asian | D |
| Grover [16] | 18 | HLA-B*15:02 | Valproate | SJS, TEN | T | Asian | E |
| Grover [16] | 19 | HLA-B*15:02 | Phenobarbital | SJS, TEN | T | Asian | F |
| Grover [16] | 20 | HLA-B*15:02 | Levetiracetam | SJS, TEN | T | Asian | G |
| Grover [16] | 16 | HLA-B*15:02 | Phenytoin | SJS, TEN | T | Asian | H |
| Li [18] | 21 | HLA-B*15:02 | Phenytoin | SJS, TEN | T | Asian | H |
| Deng [33] | 5 | HLA-B*15:02 | Lamotrigine | MPE | T | Asian | I |
| Grover [16] | 11 | HLA-B*15:02 | Carbamazepine | HSS, MPE | T | Asian | J |
| Deng [33] | 6 | HLA-B*24:02 | Lamotrigine | SJS, TEN | T | Asian | K |
| Deng [33] | 7 | HLA-B*24:02 | Lamotrigine | MPE | T | Asian | L |
| Deng [33] | 8 | HLA-B*33:03 | Lamotrigine | MPE | T | Asian | M |
| Deng [33] | 9 | HLA-B*58:01 | Lamotrigine | MPE | T | Asian | N |
| Wang [35] | 28 | HLA-B*58:01 | Carbamazepine | SJS, TEN | T | Asian | O |
| Grover [16] | 13 | HLA-A*31:01 | Carbamazepine | HSS, MPE | T | Asian, European and American | P |
| Grover [16] | 12 | HLA-A*31:01 | Carbamazepine | SJS, TEN | T | Asian, European and American | Q |
| Yip [37] | 32 | HLA-A*31:01 | Carbamazepine | Hypersensitivity | T | Asian and European | R |
| Wang [35] | 24 | HLA-B*40:01 | Carbamazepine | SJS, TEN | T | Asian | S |
| Wang [35] | 26 | HLA-B*15:11 | Carbamazepine | SJS, TEN | T | Asian | T |
| Wang [35] | 27 | HLA-B*46:01 | Carbamazepine | SJS, TEN | T | Asian | U |
| Wu [36] | 29 | CYP2C9*3 | Phenytoin | SJS, TEN | T | Asian | V |
| Grover [16] | 14 | HLA-B*15:02 | Carbamazepine | SJS, TEN | U | Asian | * |
| Grover [16] | 15 | HLA-B*15:02 | Carbamazepine | HSS, MPE | U | Asian | * |
| Wang [35] | 25 | HLA-B*15:11 | Carbamazepine | SJS, TEN | U | Asian | * |
| Wu [36] | 30 | CYP2C9*3 | Phenytoin | SJS, TEN | U | Asian | * |

On the right: author's column of the nine included systematic review; identification number (IdP) of the 33 published meta-analysis. On the left: association's column: the 22 unique associations of the same triplet [genotype -drug-ADR]. Rows are ordered by alphabetical order of the association's column. ADR: adverse drug reaction; HSS: hypersensitive syndrome; MPE: Maculopapular exanthema, SCRs: Serious Cutaneous Reactions, SJS: Stevens—Johnson syndrome, TEN: Toxic Epidermal Necrolysis. Control group: treated tolerant patients (T) and/or untreated patients (U).

*: the comparisons using untreated patients as control were not included in the association's analysis

The two significant publication bias identified by the asymmetry tests affected the associations of HLA-B*15:02 and carbamazepine induced SCRs (Begg's test, p-value = 0.02, see Fig 2) and SJS and TEN related to phenytoin (Egger's test, p = 0.03, see Fig 3).

## Exploration of the impact of publication bias

The Trim and Fill estimates were calculable for 7 out the 22 funnel plots. Most of the associations suggested an increased risk of antiseizure drug's AE with the pharmacogenetic

**Table 2. Prevalence of the publication bias.**

| Association | Genotype | Drug | ADR | Number of studies | Funnel plots analyses | | |
|---|---|---|---|---|---|---|---|
| | | | | | Visual inspection | Egger's p value | Begg's p value |
| A | HLA-B*15:02 | Carbamazepine | SCRs | 9 | Likely | 0.22 | 0.02* |
| B | HLA-B*15:02 | Carbamazepine | SJS | 4 | ND | NA | NA |
| C | HLA-B*15:02 | Carbamazepine | SJS, TEN | 18 | Likely | 0.12 | 0.24 |
| D | HLA-B*15:02 | Lamotrigine | SJS, TEN | 7 | Likely | 0.06 | 0.22 |
| E | HLA-B*15:02 | Valproate | SJS, TEN | 1 | ND | NA | NA |
| F | HLA-B*15:02 | Phenobarbital | SJS, TEN | 1 | ND | NA | NA |
| G | HLA-B*15:02 | Levetiracetam | SJS, TEN | 1 | ND | NA | NA |
| H | HLA-B*15:02 | Phenytoin | SJS, TEN | 5 | Likely | 0.03* | 0.14 |
| I | HLA-B*15:02 | Lamotrigine | MPE | 6 | Unlikely | 0.94 | 0.57 |
| J | HLA-B*15:02 | Carbamazepine | HSS, MPE | 5 | Likely | 0.97 | 0.62 |
| K | HLA-B*24:02 | Lamotrigine | SJS, TEN | 2 | ND | NA | NA |
| L | HLA-B*24:02 | Lamotrigine | MPE | 2 | ND | NA | NA |
| M | HLA-B*33:03 | Lamotrigine | MPE | 2 | ND | NA | NA |
| N | HLA-B*58:01 | Lamotrigine | MPE | 3 | ND | NA | NA |
| O | HLA-B*58:01 | Carbamazepine | SJS, TEN | 5 | ND | NA | 0.62 |
| P | HLA-A*31:01 | Carbamazepine | HSS, MPE | 6 | ND | 0.35 | 0.57 |
| Q | HLA-A*31:01 | Carbamazepine | SJS, TEN | 6 | Unlikely | 0.41 | 0.57 |
| R | HLA-A*31:01 | Carbamazepine | Hypersensitivity | 4 | ND | NA | NA |
| S | HLA-B*40:01 | Carbamazepine | SJS, TEN | 6 | Unlikely | 0.24 | 0.19 |
| T | HLA-B*15:11 | Carbamazepine | SJS, TEN | 2 | ND | NA | NA |
| U | HLA-B*46:01 | Carbamazepine | SJS, TEN | 3 | ND | NA | NA |
| V | CYP2C9*3 | Phenytoin | SJS, TEN | 3 | ND | NA | NA |

Association: identification letter of the association ($_{[genotype-drug-ADR]}$). ADR: adverse drug reaction (HSS: hypersensitive syndrome; MPE: Maculopapular exanthema, SCRs: Serious Cutaneous Reactions, SJS: Stevens—Johnson syndrome, TEN: Toxic Epidermal Necrolysis), NA: not available.

Funnel plots analyses: visual assessment—publication bias is 'likely', 'not determinable' (ND), "unlikely"; p-values of Egger's and of Begg's tests (* stands for p<0.05).

biomarkers. The size of the associations were highly modified when taking into account a potential publication bias; the quantitative impact ranged from halving to doubling the estimation of the association. No association was qualitatively modified by taking into account the publication bias. The interaction tests between $OR_{NP}$ and $OR_{TM}$ were not significant.. The $OR_{NP}$, $OR_{TM}$, and ROR are detailed in the Table 3.

# Discussion

## Summary of evidence

Most of the published meta-analyses on pharmacogenetic biomarkers of antiseizure drug's AE reported cutaneous complications. We showed that a publication bias was not rare in the assessment of pharmacogenetic biomarkers of antiseizure drug's AE. The visual analysis of the funnel plots showed that a publication bias might affect almost one quarter of those associations in this field. Using asymmetry tests, we showed that about 10% of those associations were subject to a significant publication bias. We showed that taking into account a potential publication bias might double or halve the estimation of the risk of antiseizure drug's AE associated with those genetic biomarkers.

Our results suggested a significant publication bias for the HLA-B*15–2 and its association with the risk of carbamazepine-induced serious cutaneous reactions and of phenytoin-induced

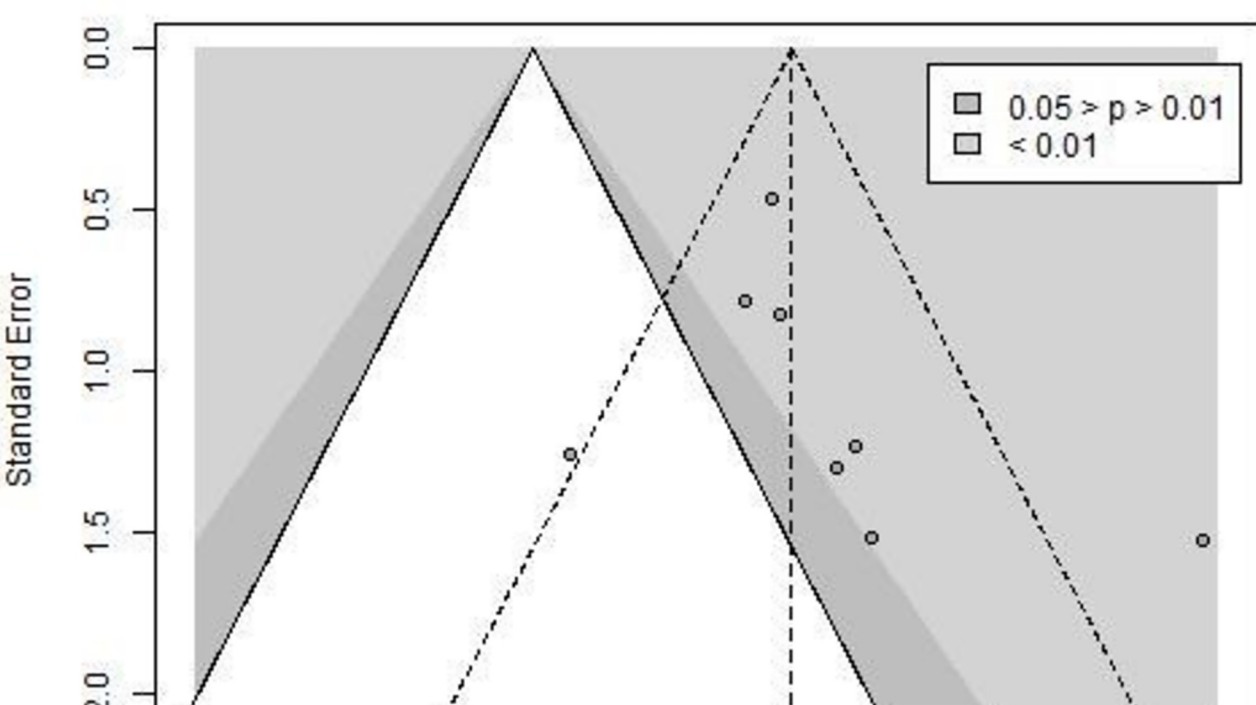

**Fig 2. Funnel plot of the association between** [HLA-B* 15:02- Carbamazepine- Serious Cutaneous Reactions (SCRs)]. Each point is a clinical studies. The white, dark, and light grey zones stand for a p value of the odds ratio i) non-significant, ii) between 0.05 and 0.01, and iii) <0.01, respectively. The dashed triangle stands for the estimation of the meta-analysis of the association, without adjusting for a potential publication bias.

SJS or TEN. This may challenge the Clinical Pharmacogenetics Implementation Consortium guidelines for carbamazepine and phenytoin [10, 11] and should be discussed in the other European pharmacogenetic networks (such as the French Network of Pharmacogenetics—RNPGx—, the Dutch Pharmacogenetics Working Group—DPWG—). However, the risk of SJS, TEN, and MPE with carbamazepine remained highly increased by the presence of the HLA-B*15:02 genotype, even when taking into account the publication bias. The effect of these associations remains high (from ≈4 to ≈40), not negating their use in clinical practice. Unfortunately, the present study was limited by the lack of power related to the study number, notably for adjusting the estimates of the association between HLA-B*15:02 and phenytoin related ADR. Finally, adjusting for publication bias affected the estimation of the pharmacogenetic associations. Indeed, even for the association between the genetic variant HLA-B*15:02 and the risk of SCRs and SJS in people treated with carbamazepine, the increase of risk appears to be overestimated by two-folds. In contrast, the association between the genetic variant HLA-A*31:01 and the risk of SJS in people treated with carbamazepine seems to be

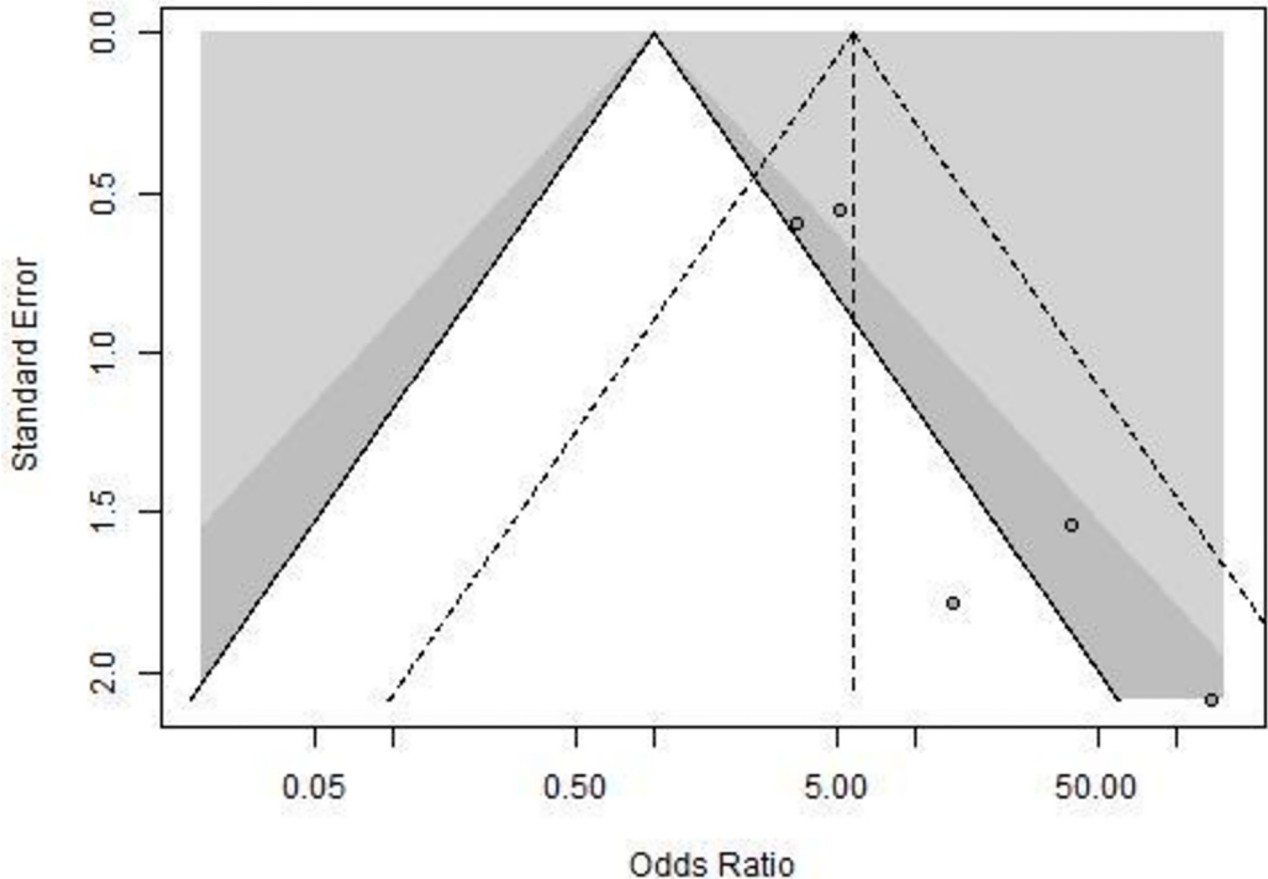

**Fig 3. Funnel plot of the association between** [HLA-B*15:02- Phenytoin- SJS/TEN (SJS: Stevens—Johnson syndrome, TEN: Toxic Epidermal Necrolysis)]. Each point is a clinical studies. The white, dark, and light grey zones stand for a p value of the odds ratio i) non-significant, ii) between 0.05 and 0.01, and iii) <0.01, respectively. The dashed triangle stands for the estimation of the meta-analysis of the association, without adjusting for a potential publication bias.

**Table 3. Exploration of the impact of publication bias.**

| Association | Genotype | Drug | ADR | k | OR$_{NP}$ | [95%CI]$_{NP}$ | OR$_{TM}$ | [95%CI]$_{TM}$ | ROR | [95%CI]$_{Inter}$ |
|---|---|---|---|---|---|---|---|---|---|---|
| A | HLA-B*15:02 | Carbamazepine | SCRs | 9 | 27.32 | [9.93; 75.17]* | 14.66 | [4.82; 44.61]* | 1.86 | [0.41; 8.39] |
| C | HLA-B*15:02 | Carbamazepine | SJS, TEN | 18 | 39.77 | [20.96; 75.48]* | 21.15 | [10.49; 42.67]* | 1.88 | [0.73; 4.86] |
| D | HLA-B*15:02 | Lamotrigine | SJS, TEN | 7 | 4.41 | [1.82; 10.68]* | 3.99 | [1.69; 9.44]* | 1.11 | [0.32; 3.80] |
| I | HLA-B*15:02 | Lamotrigine | MPE | 6 | 1.19 | [0.47; 3.00] | 1.13 | [0.47; 3.00] | 1.05 | [0.28; 3.91] |
| P | HLA-A*31:01 | Carbamazepine | HSS, MPE | 6 | 8.82 | [5.74; 13.56]* | 8.82 | [5.74; 13.56]* | 1.00 | [0.54; 1.84] |
| Q | HLA-A*31:01 | Carbamazepine | SJS, TEN | 6 | 6.46 | [2.01; 20.80]* | 12.22 | [3.39; 44.05]* | 0.53 | [0.09; 3.00] |
| S | HLA-B*40:01 | Carbamazepine | SJS, TEN | 6 | 0.28 | [0.14; 0.54]* | 0.24 | [0.11; 0.53]* | 1.17 | [0.41; 3.29] |

Association: identification letter of each association [genotype-drug-ADR]. ADR: adverse drug reaction (HSS: hypersensitive syndrome; MPE: Maculopapular exanthema, SCRs: serious cutaneous reactions, SJS: Stevens—Johnson syndrome, TEN: Toxic Epidermal Necrolysis), k: number of clinical studies involved. OR [95%CI]: odds ratio and its 95% confidence interval, Not adjusted for potential Publication bias (NP), estimated using the Trim and Fill method (TM) (when number of clinical studies >5), calculated for the interaction between not adjusted and adjusted ROR (Inter) (the ratio of ORs and its 95%CI).

* indicates 95%IC excluding the absence of association (i.e. the value 1.00)

underestimated by a factor two when taking into account a potential publication bias ($OR_{NP}$ of 6 versus $OR_{TM}$ of 12), surprisingly suggesting potential highly significant unpublished clinical studies.

## Strengths of the study

To our knowledge, our study is the first meta-epidemiological assessment of the publication bias in the pharmacogenetics of antiseizure drug adverse reactions. We used a systematic umbrella review, allowing us to gather the information from overlapping published meta-analyses and to increase the sample size. We provided an estimation of the prevalence of the publication bias in this field, using several assessments of publication bias, including independent visual analyses of the funnel plots. We also explored the impact of publication bias on the size of the effect of such association, and finally discussed the potential consequences of those results on guidelines for clinical implementation of pharmacogenetics.

## Limitations

However, our study presents several limits. First, we used the visual analysis of funnel plots, which is exposed to subjectivity. However, there is no consensual approach available for a publication bias assessment [22, 23]. This limitation illustrates the need for new tools for a publication bias assessment. Moreover, publication bias is not the only source of funnel plot asymmetry. Especially, a poor methodological quality of the small-included studies might led to the "small study effect". A true heterogeneity also might contribute to the funnel asymmetry. However, in our study, some of the punctual estimates seem to line up with usual cut-off p values for significance (see notably Fig 2, along the 1% cut off); the use of contour-enhanced funnel plots allowed to highlight that the asymmetry seems to be associated with the significance of the included studies. Moreover, we believed that the heterogeneity is limited in our funnel plots, as we kept the most precise granularity, notably by not gathering different, despite close, adverse drug reactions, in line with the author's definitions. Furthermore, tools for a publication bias assessment, as the Egger's test for example, have been initially developed for a meta-analysis of randomized trials. However, most of the available pharmacogenetic studies here were non-randomized. If the same tools for publication bias assessment may be used in meta-analysis of such non-randomized studies remains unclear. Second, despite our umbrella review approach, the number of clinical trials available remained limited for several associations. It limited the assessment of publication bias, as illustrated by the significant number of "not determinable" assessment. We were not able to provide publication bias adjusted estimates of the association between HLA-B*15:02 and phenytoin related ADR, especially. We did not conduct additional searches of original clinical studies. Indeed, focusing on published systematic review allowed assessing the impact of the publication bias in the currently available meta-analyses, which are used for guideline elaboration. We also did not remove the associations with a low number of estimates, leading to report results of little utility. Indeed, we did not consider exclusion criteria based on the number of point estimates. Therefore, we reported all the associations. Such exclusion criteria would require an arbitrary cut-off, which might be questionable. Above all, such exclusion criteria might lead to overestimate the prevalence of publication bias in the field. It would have been possible to gather together some associations, some of whom displayed similar ADRs (as "SCRs", "SJS", "SJS, TEN" considered in different associations). Indeed, some logical grouping are probably legit and relevant. This would have increased the power of the analysis. For example, combining the association A, B and C ([HLA-B*15:02 –carbamazepine–SCR/SJS/SJS,TEN] allowed to reach a nominal p value <0.05 for both the tests of Egger (p = 0.0009) and of Begg (p = 0.02) (S2 Fig in S1 File). However, this

would have probably lead to overestimate the prevalence of the publication bias in the field. Therefore, we preferred a more conservative approach by respecting the reported ADRs as defined by the authors of the clinical studies and of the published meta-analyses, despite the possible decrease of the power of our analysis. Third, the asymmetry of funnel is not entirely specific of the presence of a publication bias and can be related to heterogeneity in treatment effects. We also used a low number of studies as cut-off for using asymmetry tests. Fourth, we also used the Trim and Fill method, which requires some assumptions: it simulates potential missing studies as mirror images of observed studies [38]. Furthermore, most of the included clinical studies were at high risk of non-publication bias, including confusion and selection bias. Indeed, most of those genetic biomarker are in fact prognostic biomarkers of drug adverse reactions in a treated population. Moreover, we did not study the potential effect of subpopulations on those pharmacogenetics associations [12]. Furthermore, we used the genotype's definition reported in the published systematic review, even when rs number were not available. Last but not least, a publication bias for other associations might exist and remains invisible if most of the corresponding studies are not published at all.

## Implications of the study

Our study showed that,—as in clinical pharmacology [3] or in genetics [4]—, publication is selective in pharmacogenetics. We showed that this publication bias may affect the assessment of an association between genetic biomarker and ADR, even for consensual pharmacogenetic biomarkers of antiseizure drug's AE as HLA-B*15:02. Our study showed the need for publishing or at least registering any pharmacogenetics study, even with inconclusive results, to fight the issue of publication bias and its harmful consequences. The increased access to genomic databases and to genomic data through next generation sequencing strengthens the importance to anticipate such issues.

## Conclusion

Both the genomic area and the drug safety assessment are prone to a high risk of false positive results. Publication bias may contribute to the canonisation of such false positive associations. This may lead to not prescribing efficient drugs for false reasons, and to insufficient control of epilepsy. Moreover, false positive results dilute the true safety signal. Taking into account the publication bias is needed for correctly estimating the personalized benefit risk—balance of antiseizure drugs. Complementary to the recent "Strengthening the Reporting Of Pharmacogenetic Studies: Development of the STROPS guideline" [39], publication of negative pharmacogenetic studies is required.

## Supporting information

**S1 File.**
(DOCX)

**S1 Data.**
(7Z)

## Acknowledgments

We acknowledge Berthe El-Aya for her assistance in editing the present article.

## Author Contributions

**Conceptualization:** S. Bally, M. Cucherat, Guillaume Grenet.

**Data curation:** S. Bally.

**Formal analysis:** S. Bally.

**Investigation:** S. Bally, J. Cottin, Guillaume Grenet.

**Methodology:** M. Cucherat, Guillaume Grenet.

**Supervision:** M. Cucherat, Guillaume Grenet.

**Writing – original draft:** S. Bally, Guillaume Grenet.

**Writing – review & editing:** J. Cottin, M. C. Gagnieu, J. C. Lega, C. Verstuyft, S. Rheims, G. Lesca, M. Cucherat.

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
