## [Decision Letter · Decision Letter 0]

5 Jul 2022

PONE-D-21-20495Publication bias in pharmacogenetics of adverse reaction to antiepileptic drugs: an umbrella review and a meta-epidemiological studyPLOS ONE

Dear Dr. Grenet,

Thank you for submitting your manuscript to PLOS ONE. After careful consideration, we feel that it has merit but does not fully meet PLOS ONE’s publication criteria as it currently stands. Therefore, we invite you to submit a revised version of the manuscript that addresses the points raised during the review process.

The manuscript has been evaluated by two reviewers, and their comments are available below.

The reviewers have raised a number of major concerns. They request improvements to the reporting of methodological aspects of the study and further detail on the interpretation of some of the results.

Could you please carefully revise the manuscript to address all comments raised?

We look forward to receiving your revised manuscript.

Kind regards,

Thomas Phillips, PhD

Staff Editor

PLOS ONE

Journal Requirements:

3. Thank you for stating the following financial disclosure: "Funding

SB was funded by the UMR 5558, Université Lyon 1, CNRS (https://www.univ-lyon1.fr/).

We note that one or more of the authors is affiliated with the funding organization, indicating the funder may have had some role in the design, data collection, analysis or preparation of your manuscript for publication; in other words, the funder played an indirect role through the participation of the co-authors. If the funding organization did not play a role in the study design, data collection and analysis, decision to publish, or preparation of the manuscript and only provided financial support in the form of authors' salaries and/or research materials, please do the following:

a. Review your statements relating to the author contributions, and ensure you have specifically and accurately indicated the role(s) that these authors had in your study. These amendments should be made in the online form.

b. Confirm in your cover letter that you agree with the following statement, and we will change the online submission form on your behalf: 

“The funder provided support in the form of salaries for authors [insert relevant initials], but did not have any additional role in the study design, data collection and analysis, decision to publish, or preparation of the manuscript. The specific roles of these authors are articulated in the ‘author contributions’ section.

4. Thank you for stating the following in the Competing Interests section: "Conflict of interest 

SB, JC, MGC, GG declare that they have no competing interest. 

JCL has received speaking fees and honoraria from Roche.

CV has received consulting fees from Genzyme, Novartis and speaker honoraria from Galapagos.

SR received speaker and/or consulting fees from UCB Pharma, EISAI, GW Pharma, Idorsia, LivaNova, and Arvelle Therapeutics.

GL has received speaker honoraria from GWpharma, Eisai and Biomarin. 

MC has received consulting fees from Boehringer Ingelheim, SANOFI, AstraZeneca, EISAI and speaker honoraria from SANOFI."

We note that you received funding from a commercial sources: Roche, Genzyme, Novartis, UCB Pharma, EISAI, GW Pharma, Idorsia, LivaNova, Arvelle Therapeutics, Biomarin, Boehringer Ingelheim, SANOFI, and AstraZeneca.

Reviewers' comments:

Reviewer's Responses to Questions

**Comments to the Author**

1. Is the manuscript technically sound, and do the data support the conclusions?

Reviewer #1: Partly

Reviewer #2: Yes

2. Has the statistical analysis been performed appropriately and rigorously? 

Reviewer #1: I Don't Know

Reviewer #2: Yes

3. Have the authors made all data underlying the findings in their manuscript fully available?

Reviewer #1: Yes

Reviewer #2: Yes

4. Is the manuscript presented in an intelligible fashion and written in standard English?

Reviewer #1: No

Reviewer #2: Yes

5. Review Comments to the Author

Reviewer #1: In the manuscript "Publication bias in pharmacogenetics of adverse reaction to antiepileptic drugs: an umbrella review and a meta-epidemiological study", the authors describe an umbrella review of systematic reviews and meta-analyses of pharmacogenetic associations with adverse drug reactions to antiepileptic medications. The authors limited their review to previously published meta-analyses and systematic reviews to identify gene-drug-adverse reaction associations with multiple studies. The authors were able to identify 22 distinct associations which were analyzed for publication bias using funnel plots and tests of asymmetry. Overall, the authors make an attempt at addressing a potentially important problem when making clinical guidelines with regard to pharmacogenetic risks when taking antiepileptics. I agree with the final conclusion that studies of pharmacogenetic interactions should likely be registered even if negative in order to provide complete data. There are several concerns from this reviewer, however.

1. There are multiple grammar and word choice errors throughout the manuscript that are distracting at best but may actually affect the author's meaning or intent in certain instances at worst. A thorough, careful editing is necessary to make the manuscript more readable and ensure clear communication to the reader.

2. With regard to the author's methods, it is understandable that an umbrella review of meta-analyses and systematic reviews was utilized in order to exclude one-off pharmacogenetic interactions, but it is unclear why the authors did not conduct additional searches for specific interactions identified from the reviews in order to ensure that all publications were identified. Furthermore, if the intent was to exclude interactions with low numbers of publications, why were interactions with only one publication mentioned in a review included as identified interactions?

3. The authors state that 14 of 22 funnel plots were "not determinable" (line 231). Many of these "not determinable" funnel plots have low number of studies (table 2 and supplementary data). It was noted that all interactions with 1-4 studies were ND; 4 or less is studies is low enough to likely make funnel plot examination of little utility. It is unclear why these interactions were not simply outright excluded.

4. The reviewer is also unclear why interactions are studied so specifically. Given that there are cutaneous reactions of varying severity associated with overlapping genotypes and medications, would attempting a meta-analysis of all reactions for one variable (genotype, SJS) or combination of variable (e.g. SJS and HLA type) give better power to study publication bias in the field. On a related note, do the adverse drug reactions represent varying degrees of severity on a spectrum of cutaneous reactions or are the less severe reactions mediated by different mechanism and therefore have a specific dermatologic/histologic diagnosis? If they are all a similar mechanism, this would certainly allow for some logical grouping (e.g. all reactions in individuals with HLA-B*15:02 taking carbamazepine).

5. It is my understanding regarding the Egger's test that it is appropriate for randomized controlled trials but it is unclear if it is appropriate for studying pharmacogenetic interactions. The authors briefly mention in limitations that there are no established methods for studying publication bias in pharmacogenetics but this point is not specifically addressed.

Reviewer #2: This is very interesting paper addressing very important issue on publication bias in pharmacogenetic biomarkers (PGx) and antiepileptics adverse drug reactions (ADRs). This is very important topic and yet, have not received very much attention. The authors have confirmed that publication bias in the pharmacogenetic of antiepileptic ADRs is not uncommon and may affect the estimation of the effect of such biomarkers. I have a minor question that needs a clarification.

The results from this present study suggested that there is a significant publication bias for the HLA-B*15-2 and its association with the risk of carbamazepine-induced SCARs. Could you please clarify how to interpret this bias despite the fact that the use of HLA-B*15-02 screening has led to dramatically decrease in the incidence of SJS/TEN in Southeast Asia? How should we use this data in clinical practice?

6. PLOS authors have the option to publish the peer review history of their article (what does this mean?). If published, this will include your full peer review and any attached files.

Reviewer #1: No

Reviewer #2: No

---

## [Author Response · Author response to Decision Letter 0]

10 Oct 2022

[For clarity purpose, you'll find the respons in a specific .doc file, including special text formatting]

PLOS ONE Decision: Revision required [PONE-D-21-20495] - [EMID:04cdbb83f48c962a]

Response to Reviewers

Dear Editor, Dear Reviewers, 

Thank you very much for your time and advices. Please find below our corrections following your comments. 

Of note, we realized that the initial calculation of the ratio of ORs, the ROR, omitted to take into account the variance of the adjusted ORs (ORTM). We corrected the corresponding results section “Exploration of the impact of publication bias” by removing the sentence “The interaction test for the association of [HLA-B*15:02-carbamazepine-SJS/TEN] (identification letter #C in Table 3) almost reach significativity (ORNP=39.77 [20.96; 75.48]; ORTM =21.15 [10.49; 42.67]; interaction test: ROR=1.88 [0.99; 3.57]).” and the table 3. Fortunately, this does not change the results, as the confidence intervals of the ROR were already not significant, they just became larger. We sincerely apologize for this mistake.

Best regards,

The corresponding author on behalf of the authors

Journal Requirements:

-----

ANSWER

My bad, SB has not been funded by a specific project grant, it was a reference to its legally required traineeship grant. We corrected “The author(s) received no financial support for the research, authorship, and/or publication of this article.”

-----

3. Thank you for stating the following financial disclosure: "Funding

SB was funded by the UMR 5558, Université Lyon 1, CNRS (https://www.univ-lyon1.fr/).

-----

ANSWER

My bad, there was no specific funding. We corrected “The author(s) received no financial support for the research, authorship, and/or publication of this article.”

-----

We note that one or more of the authors is affiliated with the funding organization, indicating the funder may have had some role in the design, data collection, analysis or preparation of your manuscript for publication; in other words, the funder played an indirect role through the participation of the co-authors. If the funding organization did not play a role in the study design, data collection and analysis, decision to publish, or preparation of the manuscript and only provided financial support in the form of authors' salaries and/or research materials, please do the following:

a. Review your statements relating to the author contributions, and ensure you have specifically and accurately indicated the role(s) that these authors had in your study. These amendments should be made in the online form.

b. Confirm in your cover letter that you agree with the following statement, and we will change the online submission form on your behalf: 

“The funder provided support in the form of salaries for authors [insert relevant initials], but did not have any additional role in the study design, data collection and analysis, decision to publish, or preparation of the manuscript. The specific roles of these authors are articulated in the ‘author contributions’ section.

-----

ANSWER

My bad, there was no specific funding. We corrected “The author(s) received no financial support for the research, authorship, and/or publication of this article.”

-----

4. Thank you for stating the following in the Competing Interests section: "Conflict of interest 

SB, JC, MGC, GG declare that they have no competing interest. 

JCL has received speaking fees and honoraria from Roche.

CV has received consulting fees from Genzyme, Novartis and speaker honoraria from Galapagos.

SR received speaker and/or consulting fees from UCB Pharma, EISAI, GW Pharma, Idorsia, LivaNova, and Arvelle Therapeutics.

GL has received speaker honoraria from GWpharma, Eisai and Biomarin. 

MC has received consulting fees from Boehringer Ingelheim, SANOFI, AstraZeneca, EISAI and speaker honoraria from SANOFI."

We note that you received funding from a commercial sources: Roche, Genzyme, Novartis, UCB Pharma, EISAI, GW Pharma, Idorsia, LivaNova, Arvelle Therapeutics, Biomarin, Boehringer Ingelheim, SANOFI, and AstraZeneca.

-----

ANSWER

We added the PLOS ONE sentence “I have read the journal's policy and the authors of this manuscript have the following competing interests:”

-----

-----

ANSWER

We added the sentence “This does not alter our adherence to PLOS ONE policies on sharing data and materials.”

-----

5. In your Data Availability statement, you have not specified where the minimal data set underlying the results described in your manuscript can be found. 

-----

ANSWER

The sentence “All relevant data are within the manuscript and its Supporting Information files.” Might be corrected as the following: “All relevant data are within the manuscript and its Supporting Information 4: Results of individual studies”

-----

-----

ANSWER

Done 

-----

Reviewers' comments:

5. Review Comments to the Author

Reviewer #1: In the manuscript "Publication bias in pharmacogenetics of adverse reaction to antiepileptic drugs: an umbrella review and a meta-epidemiological study", the authors describe an umbrella review of systematic reviews and meta-analyses of pharmacogenetic associations with adverse drug reactions to antiepileptic medications. The authors limited their review to previously published meta-analyses and systematic reviews to identify gene-drug-adverse reaction associations with multiple studies. The authors were able to identify 22 distinct associations which were analyzed for publication bias using funnel plots and tests of asymmetry. Overall, the authors make an attempt at addressing a potentially important problem when making clinical guidelines with regard to pharmacogenetic risks when taking antiepileptics. I agree with the final conclusion that studies of pharmacogenetic interactions should likely be registered even if negative in order to provide complete data. There are several concerns from this reviewer, however.

1. There are multiple grammar and word choice errors throughout the manuscript that are distracting at best but may actually affect the author's meaning or intent in certain instances at worst. A thorough, careful editing is necessary to make the manuscript more readable and ensure clear communication to the reader.

-----

ANSWER

Thanks, we sent the paper to our university team for language editing. 

-----

2. With regard to the author's methods, it is understandable that an umbrella review of meta-analyses and systematic reviews was utilized in order to exclude one-off pharmacogenetic interactions, but it is unclear why the authors did not conduct additional searches for specific interactions identified from the reviews in order to ensure that all publications were identified. Furthermore, if the intent was to exclude interactions with low numbers of publications, why were interactions with only one publication mentioned in a review included as identified interactions?

-----

ANSWER

Thanks, we added: 

“We did not conduct additional searches of original clinical studies. Indeed, focusing on published systematic review allow assessing the impact of the publication bias in the currently available meta-analyses, which are used for guidelines elaboration. ”

And please see answer to question 3

-----

3. The authors state that 14 of 22 funnel plots were "not determinable" (line 231). Many of these "not determinable" funnel plots have low number of studies (table 2 and supplementary data). It was noted that all interactions with 1-4 studies were ND; 4 or less is studies is low enough to likely make funnel plot examination of little utility. It is unclear why these interactions were not simply outright excluded.

-----

ANSWER

Thanks, we added: 

“We also did not remove the associations with a low number of estimates, leading to report results of little utility. Indeed, we did not consider exclusion criteria based on the number of point estimates. Therefore, we reported all the associations. Such exclusion criteria would require an arbitrary cut-off, which might be questionable. Above all, such exclusion criteria might lead to overestimate the prevalence of publication bias in the field. ”

-----

4. The reviewer is also unclear why interactions are studied so specifically. Given that there are cutaneous reactions of varying severity associated with overlapping genotypes and medications, would attempting a meta-analysis of all reactions for one variable (genotype, SJS) or combination of variable (e.g. SJS and HLA type) give better power to study publication bias in the field. On a related note, do the adverse drug reactions represent varying degrees of severity on a spectrum of cutaneous reactions or are the less severe reactions mediated by different mechanism and therefore have a specific dermatologic/histologic diagnosis? If they are all a similar mechanism, this would certainly allow for some logical grouping (e.g. all reactions in individuals with HLA-B*15:02 taking carbamazepine).

-----

ANSWER

Thanks, we added: 

“Indeed, some logical grouping are probably legit and relevant. This would have increase the power of the analysis. For example, combining the association A, B and C ([HLA-B*15:02 – carbamazepine – SCR/SJS/SJS,TEN] allowed to reach a nominal p value <0.05 for both the tests of Egger (p = 0.0009) and of Begg (p = 0.02) (supplementary figure S2). However, this would have probably lead to overestimate the prevalence of the publication bias in the field. Therefore,… ”

-----

5. It is my understanding regarding the Egger's test that it is appropriate for randomized controlled trials but it is unclear if it is appropriate for studying pharmacogenetic interactions. The authors briefly mention in limitations that there are no established methods for studying publication bias in pharmacogenetics but this point is not specifically addressed.

-----

ANSWER

Thanks, we added: 

“Moreover, tools for publications bias assessment, as the Egger's test for example, have been initially developed for meta-analysis of randomized trials. However, most of the available pharmacogenetic studies here were non-randomized. If the same tools for publications bias assessment may be used in meta-analysis of such non-randomized studies remains unclear”

-----

Reviewer #2: This is very interesting paper addressing very important issue on publication bias in pharmacogenetic biomarkers (PGx) and antiepileptics adverse drug reactions (ADRs). This is very important topic and yet, have not received very much attention. The authors have confirmed that publication bias in the pharmacogenetic of antiepileptic ADRs is not uncommon and may affect the estimation of the effect of such biomarkers. I have a minor question that needs a clarification.

The results from this present study suggested that there is a significant publication bias for the HLA-B*15-2 and its association with the risk of carbamazepine-induced SCARs. Could you please clarify how to interpret this bias despite the fact that the use of HLA-B*15-02 screening has led to dramatically decrease in the incidence of SJS/TEN in Southeast Asia? How should we use this data in clinical practice? 

-----

ANSWER

Thanks, we added: 

“The effect size of these associations remain high (from ≈4 to ≈40), not negating their use in clinical practice. ” 

-----

---

## [Decision Letter · Decision Letter 1]

10 Nov 2022

PONE-D-21-20495R1Publication bias in pharmacogenetics of adverse reaction to antiseizure drugs: an umbrella review and a meta-epidemiological studyPLOS ONE

Dear Dr. Grenet,

Thank you for submitting your manuscript to PLOS ONE. After careful consideration, we feel that it has merit but does not fully meet PLOS ONE’s publication criteria as it currently stands. Therefore, we invite you to submit a revised version of the manuscript that addresses the points raised during the review process.

We look forward to receiving your revised manuscript.

Kind regards,

Huijuan Cao, Ph.D.

Academic Editor

PLOS ONE

Journal Requirements:

Reviewers' comments:

Reviewer's Responses to Questions

**Comments to the Author**

1. If the authors have adequately addressed your comments raised in a previous round of review and you feel that this manuscript is now acceptable for publication, you may indicate that here to bypass the “Comments to the Author” section, enter your conflict of interest statement in the “Confidential to Editor” section, and submit your "Accept" recommendation.

Reviewer #1: All comments have been addressed

Reviewer #3: (No Response)

2. Is the manuscript technically sound, and do the data support the conclusions?

Reviewer #1: Yes

Reviewer #3: Yes

3. Has the statistical analysis been performed appropriately and rigorously? 

Reviewer #1: Yes

Reviewer #3: Yes

4. Have the authors made all data underlying the findings in their manuscript fully available?

Reviewer #1: Yes

Reviewer #3: Yes

5. Is the manuscript presented in an intelligible fashion and written in standard English?

Reviewer #1: Yes

Reviewer #3: Yes

6. Review Comments to the Author

Reviewer #1: (No Response)

Reviewer #3: In the manuscript entitled "Publication bias in pharmacogenetics of adverse reaction to antiseizure drugs: an umbrella review and a meta-epidemiological study", the author analysed the error estimation of the correlation between pharmacogenetic biomarkers and antiseizure drug's adverse effects based on the discussion of publication bias. This is a very interesting article, and the research idea is very creative. After a revision of this manuscript, there is no obvious loophole in the article. Author's conclusions drawn seemed to be unbiased with rigorous methodology, therefore, I just give some minor comments on it.

1) The author only selected Medline database in the retrieval process, but did not seem to explain the specific reasons. I am afraid that some literature may be missing. Although the author uses other methods to supplement the search results except database retrieval, I believe that multiple relevant databases should be used in the retrieval process to avoid omission.

2) The Egger and Begg's methods are the most commonly used methods for funnel chart symmetry test, but publication bias is not the only source of asymmetry, such as methodological design and statistical analysis methods may affect it. Although the author mentioned this in the chapter of research limitations, I want to know the author's views on this issue, because in my opinion, this issue can reduce its impact on the results through some methods.

This question just makes me curious, because I have consulted some methodological articles, and the relevant items mentioned in them are not applicable to the author's research methods. I just made some guesses about whether subgroup analysis or meta regression can solve this problem.

7. PLOS authors have the option to publish the peer review history of their article (what does this mean?). If published, this will include your full peer review and any attached files.

Reviewer #1: No

Reviewer #3: **Yes: **Yi Yuan

---

## [Author Response · Author response to Decision Letter 1]

18 Nov 2022

PONE-D-21-20495R1_rev2

Dear Editor, Dear Reviewers, 

Thank you very much for your time and advices. Please find below our corrections following your comments. 

Best regards,

The corresponding author on behalf of the authors

6. Review Comments to the Author

Reviewer #1: (No Response)

-----

ANSWER: NA

-----

Reviewer #3: In the manuscript entitled "Publication bias in pharmacogenetics of adverse reaction to antiseizure drugs: an umbrella review and a meta-epidemiological study", the author analysed the error estimation of the correlation between pharmacogenetic biomarkers and antiseizure drug's adverse effects based on the discussion of publication bias. This is a very interesting article, and the research idea is very creative. After a revision of this manuscript, there is no obvious loophole in the article. Author's conclusions drawn seemed to be unbiased with rigorous methodology, therefore, I just give some minor comments on it.

-----

ANSWER: thank you very much!

-----

1) The author only selected Medline database in the retrieval process, but did not seem to explain the specific reasons. I am afraid that some literature may be missing. Although the author uses other methods to supplement the search results except database retrieval, I believe that multiple relevant databases should be used in the retrieval process to avoid omission.

-----

ANSWER: Indeed, we added the following in the methods section: 

“We limited the information source to the Medline database. Indeed, the Clinical Pharmacogenetics Implementation Consortium (CPIC) Guideline for HLA genotype and Use of Carbamazepine and Oxcarbazepine was restricted to the use of “the PubMed database”11. Using the same restriction allowed to estimate the impact of publication bias in the context of the current practice of guidelines elaboration in this area.”

-----

2) The Egger and Begg's methods are the most commonly used methods for funnel chart symmetry test, but publication bias is not the only source of asymmetry, such as methodological design and statistical analysis methods may affect it. Although the author mentioned this in the chapter of research limitations, I want to know the author's views on this issue, because in my opinion, this issue can reduce its impact on the results through some methods.

This question just makes me curious, because I have consulted some methodological articles, and the relevant items mentioned in them are not applicable to the author's research methods. I just made some guesses about whether subgroup analysis or meta regression can solve this problem.

-----

ANSWER: Indeed, we added the following in the discussion section: 

“Moreover, publication bias is not the only source of funnel plot asymmetry. Especially, a poor methodological quality of the small-included studies might led to the “small study effect”. A true heterogeneity also might contribute to the funnel asymmetry. However, in our study, some of the punctual estimates seem to line up with usual cut-off p values for significance (see notably figure 2, along the 1% cut off); the use of contour-enhanced funnel plots allowed to highlight that the asymmetry seems to be associated with the significance of the included studies. Moreover, we believed that the heterogeneity is limited in our funnel plots, as we kept the most precise granularity, notably by not gathering different, despite close, adverse drug reactions, in line with the author’s definitions.”

 -----

---

## [Decision Letter · Decision Letter 2]

24 Nov 2022

Publication bias in pharmacogenetics of adverse reaction to antiseizure drugs: an umbrella review and a meta-epidemiological study

PONE-D-21-20495R2

Dear Dr. Grenet,

We’re pleased to inform you that your manuscript has been judged scientifically suitable for publication and will be formally accepted for publication once it meets all outstanding technical requirements.

Kind regards,

Huijuan Cao, Ph.D.

Academic Editor

PLOS ONE

Additional Editor Comments (optional):

Reviewers' comments:

Reviewer's Responses to Questions

**Comments to the Author**

1. If the authors have adequately addressed your comments raised in a previous round of review and you feel that this manuscript is now acceptable for publication, you may indicate that here to bypass the “Comments to the Author” section, enter your conflict of interest statement in the “Confidential to Editor” section, and submit your "Accept" recommendation.

Reviewer #3: All comments have been addressed

2. Is the manuscript technically sound, and do the data support the conclusions?

Reviewer #3: Yes

3. Has the statistical analysis been performed appropriately and rigorously? 

Reviewer #3: Yes

4. Have the authors made all data underlying the findings in their manuscript fully available?

Reviewer #3: Yes

5. Is the manuscript presented in an intelligible fashion and written in standard English?

Reviewer #3: Yes

6. Review Comments to the Author

Reviewer #3: (No Response)

7. PLOS authors have the option to publish the peer review history of their article (what does this mean?). If published, this will include your full peer review and any attached files.

Reviewer #3: **Yes: **Yi Yuan

---

## [Editor Report · Acceptance letter]

20 Dec 2022

PONE-D-21-20495R2 

Publication bias in pharmacogenetics of adverse reaction to antiseizure drugs: an umbrella review and a meta-epidemiological study 

Dear Dr. Grenet:

I'm pleased to inform you that your manuscript has been deemed suitable for publication in PLOS ONE. Congratulations! Your manuscript is now with our production department. 

Kind regards, 

on behalf of

Dr. Huijuan Cao 

Academic Editor

PLOS ONE